# Paired Primary and Recurrent Rhabdoid Meningiomas: Cytogenetic Alterations, *BAP1* Gene Expression Profile and Patient Outcome

**DOI:** 10.3390/biology13050350

**Published:** 2024-05-16

**Authors:** Patricia Alejandra Garrido Ruiz, Álvaro Otero Rodriguez, Luis Antonio Corchete, Victoria Zelaya Huerta, Alejandro Pasco Peña, Cristina Caballero Martínez, Joaquín González-Carreró Fojón, Inmaculada Catalina Fernández, Juan Carlos López Duque, Laura Zaldumbide Dueñas, Lorena Mosteiro González, María Aurora Astudillo, Aurelio Hernández-Laín, Emma Natalia Camacho Urkaray, María Amparo Viguri Diaz, Alberto Orfao, María Dolores Tabernero

**Affiliations:** 1Neurosurgery Service of the University Hospital of Salamanca, 37007 Salamanca, Spain; pagarrido@usal.es (P.A.G.R.); aoteror@saludcastillayleon.es (Á.O.R.); 2Instituto de Investigación Biomédica de Salamanca (IBSAL), 37007 Salamanca, Spain; lacorsan@usal.es (L.A.C.); orfao@usal.es (A.O.); 3Pathology Service of the University Hospital of Pamplona, 31008 Pamplona, Spain; mv.zelaya.huerta@navarra.es (V.Z.H.); aa.pasco.pena@navarra.es (A.P.P.); mc.caballero.martinez@navarra.es (C.C.M.); 4Pathology Service of the Hospital Álvaro Cunqueiro, 36312 Vigo, Spain; jgonfoj@sergas.es; 5Pathology Service of the Hospital Puerta del Mar, 11009 Cádiz, Spain; mycafe@telefonica.net; 6Pathology Service of the Hospital of Basurto, 48013 Bilbao, Spain; juancarlos.lopezduque@osakidetza.eus; 7Pathology Service of the University Hospital Cruces, 48903 Barakaldo, Spain; laura.zaldumbideduenas@osakidetza.eus (L.Z.D.); lorena.mosteirogonzalez@osakidetza.eus (L.M.G.); 8Biobanco del Principado de Asturias (BBPA), 33011 Oviedo, Spain; biobanco@ispasturias.es; 9Pathology Service of the University Hospital 12 Octubre, Universidad Complutense, 28041 Madrid, Spain; aurelio.hlain@salud.madrid.org; 10University Hospital Araba, 01009 Gastei, Spain; emmanatalia.camachourkaray@osakidetza.eus (E.N.C.U.); mariaamparo.viguridiaz@osakidetza.eus (M.A.V.D.); 11Centre for Cancer Research (CIC-IBMCC; CSIC/USAL; IBSAL) and Department of Medicine, University of Salamanca, 37007 Salamanca, Spain; 12Biomedical Research Networking Centre on Cancer–CIBERONC (CB16/12/00400), Institute of Health Carlos III, 37007 Salamanca, Spain

**Keywords:** rhabdoid meningioma, primary tumor, recurrence, genetic instability, *BAP1*, prognosis, follow-up, chromosomal changes

## Abstract

**Simple Summary:**

Rhabdoid meningiomas are a rare subtype of (per definition) grade 3 meningiomas of unknown cause, with a heterogeneous clinical course and higher recurrence rates, even among tumors undergoing complete surgical removal, for whom early postoperative radiotherapy may favor local control of the disease and prolonged survival. Here, we compared the clinical, biological and genetic features (at diagnosis and in sequential tumor recurrences) of recurrent vs. non-recurrent rhabdoid meningiomas in a retrospective series of 15 patients with a long follow-up. Recurrent RM showed a higher genetic instability at diagnosis associated with multiple chromosomal losses involving chromosomes 1p, 14q, 18 and 22. Non-recurrent RM were genetically less complex tumors which more frequently showed extensive losses at chromosome 19p and/or 19q, together with gains of chromosomes 20 and 21. Comparison of paired primary vs. recurrent tumor samples of recurrent RM revealed additional losses of chromosomes 16q and 19p, together with gains of chromosomes 1q and 17q. Of note, focal chromosome gains at 17q22 were exclusively found in women with recurrent RM who died, although survival differences did not reach statistical significance.

**Abstract:**

Rhabdoid meningiomas (RM) are a rare meningioma subtype with a heterogeneous clinical course which is more frequently associated with recurrence, even among tumors undergoing-complete surgical removal. Here, we retrospectively analyzed the clinical-histopathological and cytogenetic features of 29 tumors, from patients with recurrent (seven primary and 14 recurrent tumors) vs. non-recurrent RM (*n* = 8). Recurrent RM showed one (29%), two (29%) or three (42%) recurrences. *BAP1* loss of expression was found in one third of all RM at diagnosis and increased to 100% in subsequent tumor recurrences. Despite both recurrent and non-recurrent RM shared chromosome 22 losses, non-recurrent tumors more frequently displayed extensive losses of chromosome 19p (62%) and/or 19q (50%), together with gains of chromosomes 20 and 21 (38%, respectively), whereas recurrent RM (at diagnosis) displayed more complex genotypic profiles with extensive losses of chromosomes 1p, 14q, 18p, 18q (67% each) and 21p (50%), together with focal gains at chromosome 17q22 (67%). Compared to paired primary tumors, recurrent RM samples revealed additional losses at chromosomes 16q and 19p (50% each), together with gains at chromosomes 1q and 17q in most recurrent tumors (67%, each). All deceased recurrent RM patients corresponded to women with chromosome 17q gains, although no statistical significant differences were found vs. the other RM patients.

## 1. Introduction

Rhabdoid meningiomas (RM) are a rare subtype of meningioma defined by the presence of rhabdoid cells and graded according to the World Health Organization (WHO) classification of central nervous system tumors as WHO grade 3 tumors [1,2]. However, some authors suggest that meningiomas with rhabdoid features should be graded similarly to non-rhabdoid meningiomas [3], since the rhabdoid component (with a variable number of rhabdoid cells) can emerge already at first diagnosis in grade 1/2 tumors, or at recurrence with an increased proportion of rhabdoid cells [4,5]. Variable results have been reported about the clinical behavior and the outcome of meningioma in general [6], and of RM patients in particular [3,5]. Overall, a more adverse prognosis [7,8,9] associated with variable recurrence and death rates has been reported for RM compared to other meningiomas [4,10,11,12], pointing out the need for specific predictive biomarkers for the former tumors. In this regard, the degree of surgical resection evaluated via the Simpson classification as a gross total resection (GTR) vs. a subtotal resection (STR) grading scale based on postoperative magnetic resonance imaging (MRI), is a major universal prognostic factor in meningiomas in general and also in rhabdoid tumors. Tumor resection might be influenced by the tumor location, the extent of invasion, the attachment to vital intracranial structures, and the surgeon’s assessment during surgery [13,14]. Because of this, new grading systems for meningioma resection have been recently proposed [15] to better evaluate the effectiveness of tumor resection based on an updated Simpson system. Whether the persistence of residual tumor cells after STR [16] increases the risk of tumor regrowth [13,17,18], or whether tumors which underwent surgery in combination with radiotherapy (RT) have different recurrence rates and should be analyzed separately, still remains a matter of debate, since some patients recover after surgery alone, while others that received postsurgical RT do relapse [19].

Overall, the 5-year recurrence rates of WHO high-grade meningiomas undergoing GTR ranges between 9% and 50% [20], with median 5-year recurrence-free survival (RFS) rates of 68% [21,22]. Some patients die following disease progression or recurrence [23,24,25] with a median 5-year overall survival (OS) rate of 76% for grade 2 and of 55% for grade 3 meningiomas [19]. Despite a greater tendency of RM toward local recurrence [10,26], and to a lesser extent, even to distant metastasis [9,27], the mechanisms leading to tumor relapse remain to be elucidated. Nevertheless, several factors appear to be involved which include both the grade of tumor resection and the use of fractionated RT [28]. Thus, adjuvant RT has been recommended after surgery to reduce recurrence rates [13,19] whenever the tumor underwent STR, or even when GTR had been achieved [12,29]. Despite these recommendations, some studies suggest that subsequent complementary radiation did not have a significant benefit in RM [11], their risk of relapse remaining hard to predict, which is due in part to the highly variable follow-up times (ranging from months to several years) reported for the different series referred to in the literature [3,4,5,10,11,12]. Altogether, these findings highlight the need to increase our knowledge about RM, and specifically, about those factors involved in the recurrence of this subtype of meningioma.

Among other tumor-associated features, WHO grade 2 and 3 meningiomas emerge as those tumor subtypes which are associated with a more aggressive clinical course [1] and an increased risk of recurrence [19] under different therapeutic settings [30]. In addition, the tumor size, male sex and a younger age [31,32], together with the number and type of cytogenetic alterations [33,34], have also been identified as potential prognostic factors, and used to build different risk scores for predicting outcome in meningiomas [31,35]. Among other genetic variables, mutation-induced inactivation of the BRCA1-associated protein 1 (*BAP1*) (a tumor suppressor gene involved in regulation of gene transcription, cell cycle and growth, response to DNA damage and chromatin dynamics), has been related to a significantly decreased time to recurrence in RM [36]. Of note, *BAP1* expression has been reported to be lost in several cancer types [37], including a subset of clinically aggressive meningiomas that display a rhabdoid or papillary histology, in association with both a higher tumor grade and a significantly decreased time to recurrence [37,38]. Moreover, loss of *BAP1* gene expression also appears to be involved in familial meningiomas [39,40].

Due to the relatively high rate of recurrence of RM and the current lack of predictive factors for tumor recurrence that might help to better define the patient prognosis and determine the need for adjuvant treatment after surgery, there is an urgent need to conduct in-depth studies into this rare subtype of meningioma aimed at identifying new prognostic biomarkers through e.g., direct comparison of recurrent vs. non-recurrent tumors. To the best of our knowledge, no RM series have been reported so far in the literature in which the relationship between primary and recurrent RM has been investigated in detail in order to identify biological differences between both groups of tumors that could contribute to a better understanding of disease recurrence. In this retrospective study, we compared the underlying cytogenetic alterations of seven primary vs. 14 recurrent tumors (from a total of seven recurrent RM patients) vs. eight primary non-recurrent grade 3 RM patients. Our ultimate goal was to identify those unique genetic profiles of RM that might be associated with both the clinical course of the disease and patient outcome.

## 2. Materials and Methods

### 2.1. RM Patients and Samples

A total of 29 RM specimens from 15 patients were retrospectively collected from 9 hospitals in Spain. In total, 21 specimens collected from 7 recurrent RM patients and 8 specimens from non-recurrent RM (all grade 3), were studied. Patient selection criteria included patients with histopathological diagnosis of RM either at disease onset or at recurrence of any histopathological grade which had left-over stored tumor tissues. Recurrent RM included 4 male and 3 female patients. At diagnosis, 3/7 recurrent tumors were grade 1, one was grade 2, and the other 3 corresponded to de novo grade 3 RM, with a median (range) follow-up of 13 years (7 to 22 years). One of the 7 primary tumor samples was discarded because the decalcification process resulted in a poor-quality sample, which could not be assessed via immunohistochemistry (IHC). In addition to the primary tumors, a total of 14 recurrent RM specimens were studied for the recurrent RM patient group, which included one recurrent tumor from 2 patients (*n* = 2 specimens), 2 recurrent tumors from another 2 patients (*n* = 3 specimens; in one patient treated with bevacizumab at his second relapse in the absence of surgery and RT, the second relapse sample was not available) and 3 recurrent tumor specimens from 3 patients (*n* = 9 specimens). The second group of non-recurrent RM patients consisted of 6 male and 2 female patients, all having WHO grade 3 RM (Appendix A). No patients in our series reported a family history of meningioma. The demographics, clinical and biological features observed at diagnosis, together with the extent of the surgical resection, the use of postoperative RT, and follow-up data, including RFS (determined as the time from tumor surgery to disease recurrence or the last follow-up visit) and OS (calculated as the time from surgery to death or to the last follow-up visit), were retrospectively collected for each patient from their medical records. Formalin-fixed and paraffin-embedded (FFPE) tumor tissues were processed according to standard protocols and they were used for the different analytical procedures that are described below.

### 2.2. Molecular Analysis via Copy Number Arrays

Genetic studies were performed on retrospectively collected FFPE blocks after they had been cut into 3-micron sections and genomic DNA had been extracted using the QIAamp DNA FFPE Tissue Kit (Qiagen, Valencia, CA, USA), according to the manufacturer’s instructions. The concentration of double-stranded DNA was quantified using the Qubit^®^ dsDNA Assay (ThermoFisher Scientific, Waltham, MA, USA). DNA from each RM (40–80 ng), including primary and recurrent tumor tissues, was analyzed for genomic copy number alterations (CNA) using the Affymetrix OncoScan arrays in a GeneChips Fluidics Station 450 (Affymetrix, ThermoFisher Scientific, Waltham, MA, USA) and the OncoScan (stain and wash) reagents, according to the manufacturer’s instructions. All microarrays were scanned on a GeneChip scanner 3000 (Affymetrix). Data quality control (QC) was performed with the OncoScan Console 1.3 software, as per the recommendations of the manufacturer (Affymetrix). The CNA profile of individual samples was determined using normalized data via the Nexus Express for OncoScan software (version 3.1; Affymetrix). In addition, the OSCHP-TuScan data format and the tool of the Chas Console 1.3 software (Affymetrix) were used to identify CNA and calculate the percentage of altered cells and the overall ploidy status, including the percentage of cells per sample that displayed loss of heterozygosity. For these CNA studies, the Genome Reference Consortium Human Build 37 (GRCh37) was used to define the probe location. Gains were determined when the log2 ratio signal value obtained was >0.5, whereas losses were defined for values below the <0.5 cut-off. The chromosomal regions identified as displaying CNA were merged into a single genome interval whenever they were within a 0.5 Mb distance. Subsequently, they were filtered to exclude small regions (<0.5 Mb), if another similarly altered region was not present within the nearest 10 Mb sequences. The lowest *p*-value (from a segment of at least 50 kb in length) within the merged regions was used to annotate the regions with Chromosome Analysis Suite (ChAS) (Thermo Fisher Scientific, Waltham, MA, USA). Weighted log2 ratios were also obtained for each array probe using the ChAS console (v.4.2) (Affymetrix). These data were winsorized and segmented via the pcf (piecewise constant fragments) algorithm from the “copynumber” package (v.1.30.0) [41] in R (v.4.2.1, R Development Core Team, 2022 https://www.r-project.org, accessed on 25 June 2022). The minimal common regions and the recurrent broad alterations were calculated across all samples via GISTIC (v.6.15.28) [42], at a confidence level of 0.90 and a q-value threshold of 0.1. Focal alterations in less than five samples were identified using the CGHcall R package (version 2.62.0). Those CNA that overlapped with more than 50% of the copy number variations previously reported in the Database of Genomic Variants (DGV) [43], were excluded.

### 2.3. Histological and Immunohistochemical Studies

Hematoxylin- and eosin- (H&E) stained paraffin-embedded meningioma sections were microscopically assessed/reviewed by 2 experienced pathologists. The percentage of cells with rhabdoid morphology was evaluated (together with the tumor-specific histological patterns) for each individual sample. In addition, the mitotic rate per 10 high-power fields (10 HPF) was assessed to assign tumor grade according to the WHO criteria. Immunohistochemical staining for epithelial membrane antigen (EMA) and intracellular (cytoplasmatic) staining for vimentin and the (nuclear) Ki-67 antigen (detected via the MIB1 monoclonal antibody) was also performed on FFPE tissue sections, following standard operating procedures. In a subset of 27 available specimens (6 primary tumor samples from recurrent RM, 14 recurrent tumor samples and 7 non-recurrent RM), IHC assessment of *BAP1* protein expression using the *BAP1* (C4):sc-28383 antibody (Santa Cruz Biotechnology, Dallas, TX, USA) was performed with the Bond-III immunostainer (Leica Biosystems, Nussloch, Germany). *BAP1* expression was defined when there was homogeneous nuclear staining in ≥40% of all tumor cells, while negativity was defined when there was no nuclear staining in tumor cells compared to the appropriate internal controls, or when the percentage of negative cells was >60% of all tumor cells.

### 2.4. Statistical Methods

For all statistical analyses, the SPSS software (SPSS 25.0, IBM SPSS, Armonk, NY, USA) was used. Based on the non-parametric distribution of continuous variables, the Mann–Whitney U and the Chi-square test were used to establish the statistical significance of differences observed between different groups of patients for categorical and continuous variables, respectively. Patient RFS and OS curves were plotted according to the Kaplan–Meier method and the statistical significance of differences observed between survival curves was established via the log-rank test.

## 3. Results

### 3.1. Demographics, Clinical and Histopathological Features of Recurrent vs. Non-Recurrent RM

Upon dividing our RM patients into cases with non-recurrent vs. recurrent tumors, the following similar (*p* > 0.05) median age (range) was observed at diagnosis for both groups: 59 (34–81) vs. 61 (45–69) years. At diagnosis, the majority of the primary tumors corresponded to male patients (ten out of fifteen cases, 67%) and many of them had a high median percentage of rhabdoid cells (>50%), six out of eight (75%) non-recurrent tumors vs. two out of seven (29%) recurrent RM (*p* = 0.07), respectively (Table 1). Overall, 9/15 (60%) were WHO grade 3 meningiomas, five (33%) WHO grade 2, and one (6%) WHO grade 1 tumor. Compared to non-recurrent RM, recurrent tumors were slightly more frequently observed in women (43% vs. 25%; *p* > 0.05), they showed a tendency towards a lower frequency of rhabdoid cells (*p* = 0.07) and WHO grade 3 tumors (*p* = 0.003). Interestingly, immunohistochemical staining performed to assess *BAP1* expression at diagnosis showed positivity in around one third (69%) of all RM including seven out of seven (100%) non-recurrent RM vs. only two out of six (33%) recurrent tumors studied at diagnosis (*p* = 0.02) (Table 1). Recurrent RM showed one (29%), two (29%) or three (42%) recurrences, the rate of *BAP1* expression increasing from the first (71%) to the second (100%) and third recurrences (100%), as illustrated in Figure 1 for three recurrent RM patients. Despite the frequency of *BAP1* expression and the fact that WHO grade 3 tumors at diagnosis were significantly lower in recurrent vs. non-recurrent, it progressively increased from the primary tumors evaluated at diagnosis to the subsequent recurrent samples (Table 1).

Although GTR was achieved in a similar (*p* > 0.05) proportion of non-recurrent and recurrent RM (88% and 86%, respectively), the former more frequently received RT following diagnostic surgery (50% vs. 29%, respectively; *p* = 0.4). Despite the overall longer follow-up for the recurrent RM group (median of 13 years; range: 7–22 years) compared to the non-recurrent tumor group (median of 6 years; *p* = 0.008), the median RFS time to the first recurrence was of five years (range: 2–9 years), with recurrent tumors showing a tendency towards a higher rate of deaths (43% vs. 14%; *p* = 0.05) following a progressively lower frequency of GTR from the first recurrence to the subsequent one(s) (*p* = 0.07) (Table 2).

### 3.2. DNA Copy Number Changes in Primary Non-Recurrent vs. Recurrent RM

A high genetic instability was observed in both non-recurrent and recurrent RM. Thus, both tumor groups shared chromosome 22 losses in all (100%) RM. Despite this, non-recurrent RM showed a significantly different genetic profile associated with a lower median number of altered chromosomes, compared to recurrent tumors (Table 3). Overall, those alterations more frequently found in non-recurrent meningiomas consisted mostly of extensive losses at chromosome 19p (62%) and/or 19q (50%), together with gains of chromosomes 20 and 21 (38%, respectively) (Table 3). In turn, recurrent RM displayed broad losses of chromosomes 1p, 14q, 18p, 18q (67% each) and 21p (50%) together with focal gains at chromosome 17q22 (67%) (Table 3).

### 3.3. Pattern of Evolution of CNA Profiles in Paired Primary vs. Recurrent Tumors

The most frequent chromosomal abnormalities observed in primary recurrent RM tumor samples and their first recurrent tumors corresponded to chromosome losses (Table 4 and Figure 2) that affected large DNA regions (large deletions and/or monosomy) located at chromosomes 1p (67%), 6q (50%), 14q (67%), 18p (67%), 18q (67%), 21p (50%) and 22q (100%). However, the genetic profiles of individual recurrent tumors were heterogeneous, and only two out of seven patients (P1 and P2, both males that remain alive after 13 and 23 years of follow-up) showed identical chromosomal CNA in the primary tumor and their first recurrence (Appendix A). In the other five recurrent RM, additional chromosomal abnormalities were observed in recurrent vs. primary tumor samples (Appendix A). These consisted of additional losses of chromosomes 16q and 19p (50% each) in recurrent tumors, together with new gains involving chromosomes 1q and 17q (67%, for both alterations) (Table 4). Consequently, the overall number of genetic changes detected at each subsequent relapse increased with an average of eight affected chromosomes in patients with only one recurrence (range: 7 to 12), and 15 affected chromosomes (range: 7 to 19) in patients who had two or three relapses. Thus, several different genetic changes appeared (and less frequently disappeared) during the course of the disease. These changes included: (i) CNA of the same chromosome, with a larger or a smaller size, which even disappeared in a subsequent vs. previous recurrent tumor; (ii) CNA that affected different arms of the same chromosome usually with a larger size, such as 1q gains; and (iii) new losses and gains involving chromosomes that were not affected in the previous (recurrent) tumor of the same patient, which included alterations of chromosomes 3, 4q, 6q, 7pq, 8, 9p, 10, 12p, 15q, 16q and 20 (Appendix A). Consequently, a higher number of genes were usually involved at recurrence in the gained chromosomal 1q and 17q regions, while focal gains at 17q22 (limits: 55,403,923–57,333,858) chromosomal region were only observed at diagnosis while absent at relapse (Table 4 and Figure 3).

The overall median (range) follow-up of the whole RM patient series (at the moment of closing this study) was of 11 years (range: 3 to 22 years) with 7/15 (47%) recurrent cases, including a single relapse in two patients (one of them under early adjuvant RT after surgery), and two or three relapses in five patients who had all received RT after the first and the second surgery (*n* = 2) or only after the third surgery (*n* = 3) (Figure 3). The first (local) recurrences appeared within two to eight years from diagnostic surgery, while the time lapse to the second and third relapse was between one and ten years and between one to five years from the previous recurrence, respectively. Interestingly, the three de novo grade 3 RM had a median recurrence-free survival of seven years, compared to only four years for the lower-grade (grades 1 and 2) meningiomas with a rhabdoid component evolving to grade 3 RM at recurrence. At the moment of closing the study, three patients had died due to tumor progression, all three corresponding to those female patients who showed more complex CNA, including gains of chromosome 17q, already at diagnosis (P4, P6 and P7) (Appendix A). The percentage of deceased patients was 38% in the non-recurrent RM versus 57% in recurrent tumors, all deceased recurrent patients were females but survival differences per sex did not reach statistical significance (*p* = 0.097 vs. men) (Figure 4).

## 4. Discussion

Despite the fact that most sporadic meningiomas are considered benign tumors, a significant proportion of them still relapse after diagnostic surgery, even following radiotherapy. Of note, recurrence of sporadic meningiomas remains difficult to predict on an individual basis [20,44,45]. The specific biological mechanisms underlying an aggressive clinical course remaining unknown. Altogether, this points out the need for more accurate and robust prognostic markers particularly among patients undergoing GTR [17,18].

For decades, tumor histopathological features have been considered as the most informative criteria for both the diagnosis and the prognostic classification of meningiomas [1], RM emerging as a distinct (rare) entity potentially associated with a more aggressive clinical course and a worse patient outcome related to a greater recurrence rate [4,20,45]. In line with these findings, here we observed that around half of all RM tumors analyzed showed disease recurrences, a frequency that was by far, significantly higher than that observed among all sporadic meningiomas after a similarly long follow-up, such results supporting the poorer prognosis and outcome of RM. Interestingly, despite the fact that more than half of our patients showed primary tumors with a major rhabdoid cell component associated with WHO grade 3, in the other patients this was only observed after (a first, second or even third) tumor recurrence. Despite this, rhabdoid cells were already (systematically) detected at diagnosis in the primary WHO lower grade 1 or 2 tumor samples, where it was admixed with other histopathological components. Such increase in the rhabdoid component observed in the recurrent tumors of these later set of patients, was paralleled by an increased tumor grade towards a more atypical and anaplastic histology, as tumor recurrences developed over time. Altogether these results suggest that in those meningiomas presenting with a rhabdoid component representing less than 50% of all tumor cells and low-grade histopathological features, a good prognosis cannot be assumed, due to the potential risk of recurrence and progression to higher grade tumors, as found in our cohort. Per definition, RM are WHO grade 3 meningiomas in which rhabdoid cells might represent a major (varying) percentage of all tumor cells. Some authors have recently suggested that a rhabdoid cell morphology might represent a phenotypic variant, rather than a specific subtype of meningioma [3,11,26,34], because some RM cases that were first diagnosed as WHO grade 3 tumors were reclassified after they were reevaluated as lower grade (grade 2 or even grade 1) meningiomas with a rhabdoid cytomorphology, in the absence of anaplastic features. Overall, our results confirmed and extend these observations by showing the presence of a rhabdoid cell component in WHO grade 1 and 2 patients. However, our data also indicated that these patients might evolve into WHO grade 3 tumors with a poorer outcome even when compared to the typical primary grade 3 RM. These results support previous evidence indicating that the de novo WHO grade 3 RM might show longer progression-free survival rates than WHO grade 1 and 2 meningiomas with a rhabdoid component that evolved to WHO grade 3 RM, fully in line with our data [46,47]. Nevertheless, these data need to be confirmed in larger series due to the rarity of RM, the low number of recurrent RM analyzed in our cohort and in those reported in the literature, and the potential selection of patients with tumors with a rhabdoid component that had relapsed as a RM (excluding non-rhabdoid recurrent meningiomas). In addition, it should be noted that despite two-thirds of our patients undergoing GTR (in both recurrent and non-recurrent RM), in one third of the cases only partial tumor resection was attained and the frequency of GTR progressively decreased from diagnosis to the first and subsequent second and third recurrences, probably due to a more deteriorated patient status several years after the first surgical intervention. Interestingly, patients undergoing RT after the first surgery presented with a lower number of recurrences suggesting that patients under early adjuvant RT treatment might have prolonged RFS rates and/or time to a second relapse in association with a longer OS. However, these preliminary data are not consistent with previous findings in atypical meningiomas that indicate that early postoperative adjuvant RT has no significant impact on patient OS despite a potential benefit on RFS [48,49], the potential value of RT therapy remaining a matter of debate [44,45,50].

Altogether, the heterogeneous histopathological features, clinical course and outcome of RM patients point out the need for new biomarkers that could be of help in the diagnosis of RM and in predicting tumor behavior and disease outcome. In this regard, *BAP1* has emerged as a critical tumor suppressor gene across multiple cancer types whose mutations predispose to tumor development that might be relevant in RM [51]. Thus, among the *BAP1*-mutant families affected by the *BAP1* tumor predisposition syndrome (*BAP1*-TPDS), those individuals who have inherited a *BAP1* mutant allele may develop one or more malignancies during their lifetime, mostly uveal melanoma, malignant mesothelioma, cutaneous melanoma, renal cell carcinoma, and basal cell carcinoma [38]. Of note, hepatocellular carcinoma, cholangiocarcinoma and meningioma have also been associated with the *BAP1*-TPDS. However, these cancer types occur also sporadically in patients who carry somatic *BAP1* mutations [37]. The tumor suppressor function of *BAP1* is linked to its dual activity in the nucleus, (where it is involved in DNA repair and transcription), and in the cytoplasm (where it regulates cell death and mitochondrial metabolism). Of note, loss of *BAP1* expression due to gene inactivation by mutation has been included in the WHO classification [1] as a typical feature in high-grade rhabdoid meningiomas in association with a poorer prognosis. Although *BAP1* inactivation has been reported in patients with high-grade RM carrying germline mutations, indicating that such meningiomas can arise as part of the *BAP1*-TPDS, in other RM cases *BAP1* inactivation might be caused by somatic *BAP1* loss [36,38]. In this sense, immunostaining carried out to determine *BAP1* loss expression in our patients showed that only recurrent RM presented with loss of *BAP1* at diagnosis, while *BAP1* was positive in all the non-recurrent WHO grade 3 primary RM cases analyzed. Altogether, these results suggest a limited value of the loss of *BAP1* expression via immunohistochemistry as a biomarker of RM. These results are in line with the relatively low frequency of deletions of the chromosome 3p21.1 region (where the *BAP1* gene was coded) in our cohort, which was restricted to a few (first and second) relapsed tumor samples that still retained expression of the *BAP1* protein. Despite all the above findings, further genetic analyses are required to rule out or confirm the role of this gene in RM.

In the latest edition of the WHO classification of the Central Nervous System Tumors, as well as in multiple studies in the literature focused on meningioma, cytogenetic alterations emerge as relevant biomarkers for tumor diagnosis and prognostic stratification [35,52,53,54]. Among all, monosomy 22 is the most consistent and common cytogenetic change in meningiomas, together with other chromosomal losses and gains which are more frequently found among atypical and anaplastic vs. benign tumors [35,55]. These additional alterations include chromosome 1p deletions (associated with a poorer outcome) and losses of chromosomes 6q, 9p, 10, 14q and 18q (which occur in higher grade tumors), as well as gains of chromosomes 1q, 9q, 12q, 15q, 17q and 20q, in addition to mutations of the *NF2*, *AKM*, *TRAF7*, *SMO* and *PIK3CA* genes [31,33,56,57]. Despite this, to the best of our knowledge, no study has been reported so far in the literature which has focused on the cytogenetic analysis of paired primary and recurrent RM tumors, as performed here for the first time. Overall, our results revealed that the genetic CNA profiles of recurrent RM differed substantially from those of non-recurrent cases, despite both groups of tumors systematically sharing deletions of chromosome 22q. Thus, while non-recurrent RM showed a combination of losses of chromosome 9 (9p^−^ and 9q^−^) together with gains of chromosome 20p and 21p/21q, recurrent RM tended to show more complex CNA profiles with extensive losses of chromosomes 1p, 14q, 18 (18p and 18q) and 21p, together with focal gains at chromosome 17q22, in the absence of chromosome 19 losses and gains of chromosomes 20 and 21p/21q. Altogether, these data point out the existence of a higher genetic instability in recurrent vs. non-recurrent RM, associated with more complex karyotypes. In line with their greater genetic instability, direct comparison of the CNA (genetic) profiles of paired primary vs. recurrent tumor samples, revealed clonal evolution associated with the acquisition of additional genetic changes in the majority of the recurrent RM patients analyzed, such new CNA involving losses and gains of chromosomes 1p/q, 2q, 6q, 7p/q, 10p/q, 14q, 16q, 17p/q, 18q, 19p/q and 22q with unique patterns per patient; of note, some of the newly acquired alterations were also present in non-recurrent RM at diagnosis. In individual patients, the new CNA acquired at recurrence [3,34] included large losses and gains of several chromosomes (involving the entire chromosome, one chromosomal arm or several chromosomal bands where thousands of genes are coded), with highly variable profiles among the different patients. Despite this, in two recurrent RM, an identical CNA profile was observed in paired primary vs. recurrent tumors as previously described also in other meningiomas subtypes [35,52,54,58]. The WHO 2021 classification has proposed some specific molecular markers to be associated with grade 3 meningiomas (e.g., homozygous deletion of *CDKN2A/B*) because they confer a poorer prognosis [1]. In our cohort, only one of all recurrent RM showed homozygous deletion of the *CDKN2A/B* gene. These data are in line with previous observations that indicate that homozygous deletion of *CDKN2A/B* is infrequent in meningiomas [59,60,61]. Alterations of other relevant genes involved in meningiomas such as gains of *EGFR* (found here in a primary RM case and other recurrent RM specimens) and gains of the *MET* gene (detected here also in one patient), appear to be infrequent in RM. Apart from these genes, chromosomal regions recurrently altered in RM contained several other tumor suppressor genes and oncogenes, that might contribute to RM tumorigenesis, but whose specific role still remains to be fully understood. Of note, all patients who presented gains of chromosome 17q showed a dismal outcome, all of them being recurrent RM female patients, while the other four recurrent RM patients (all men) did not present gains of chromosome 17q and remain alive, despite tumor recurrence. However, differences in overall survival between these two groups did not reach statistical significance, which might be due to the limited number of cases analyzed and thereby, requires further investigations and confirmation in a larger series of patients with a long follow-up. In fact, the limited number of cases together with the retrospective nature of this study are important limitations that point out the need for larger prospective (collaborative) studies that avoid retrospective selection of recurrent RM patients as completed here, with left-over tumor samples to confirm our observations.

## 5. Conclusions

Here we report on the largest series of RM patients in which CNA genetic profiles were compared between primary tumors of non-recurrent vs. recurrent RM and their paired primary vs. recurrent tumor samples. Overall, our results showed a higher genetic instability at diagnosis in recurrent vs. non-recurrent RM, which involved different chromosomal regions and more complex cytogenetic profiles in recurrent vs. non-recurrent RM. Of note, in recurrent RM, clonal evolution with multiple additional chromosome gains and losses was observed at recurrence, in the majority of cases. Despite the fact that these findings might contribute to the understanding of the pathogenic mechanisms of RM, genetic associations remain challenging, due in part to the complexity of the altered profiles and the multiplicity of genes involved at relapse. Further prospective (collaborative) studies in a larger series of RM patients with long follow-up in the absence of a potential selection bias, are needed to confirm our preliminary results.

## Figures and Tables

**Figure 1 biology-13-00350-f001:**
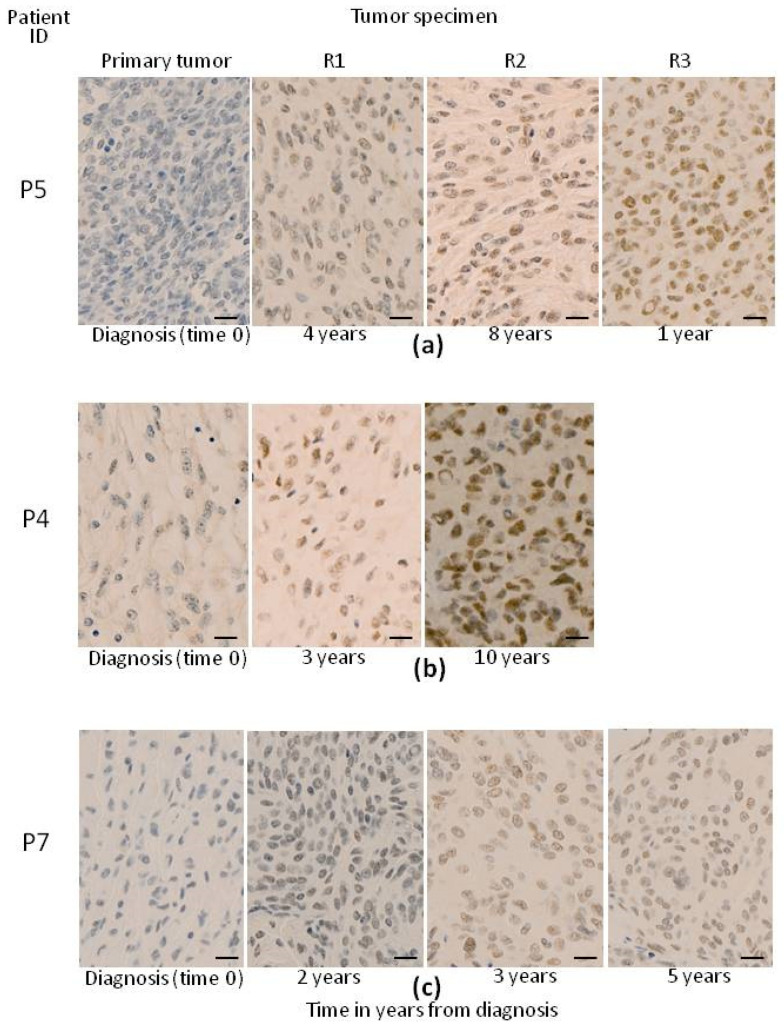
Illustrating example of the microscopic (x40 with the black scale bar indicating 50 µm) *BAP1* expression profile observed in primary tumors from three recurrent meningioma patients (P4, P5, P7) and their respective recurrent tumor specimens, as evaluated via conventional immunohistochemistry (**a**–**c**). Brown nuclear staining (strong or weak) reflects *BAP1* expression while blue nuclei (cells without *BAP1* staining) indicate cells with loss of *BAP1*.

**Figure 2 biology-13-00350-f002:**
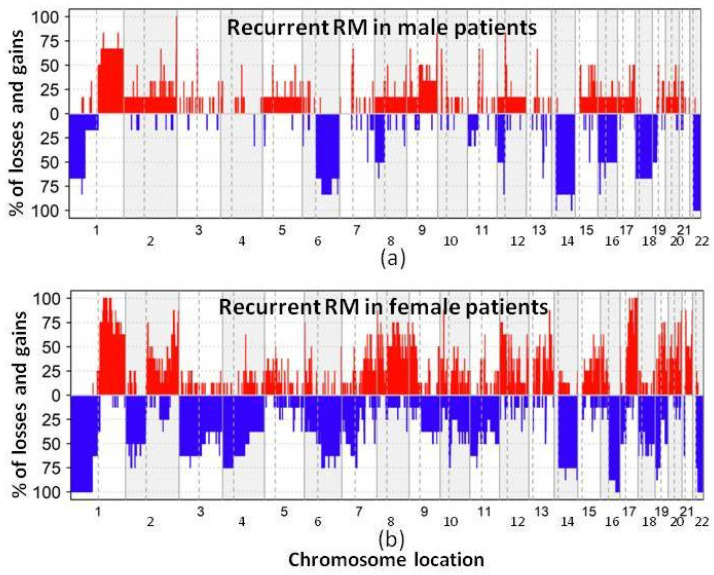
Overall frequency and localization of chromosomal CNA detected in recurrent RM from men (**a**) vs. women (**b**). Red color indicates chromosomal region gains and blue color chromosomal losses.

**Figure 3 biology-13-00350-f003:**
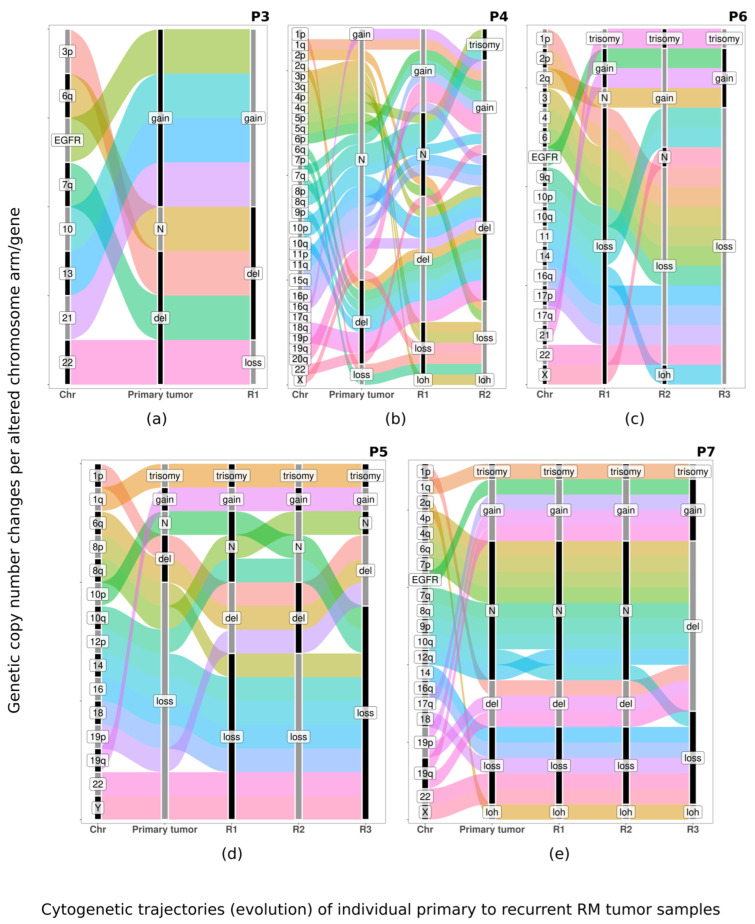
Sankey chart where in which genetic alterations (gains and losses) are represented by broad lines/bars color-coded per chromosomes. The length of individual lines/bars represents the flow of structural variants along time from diagnostic to one or multiple consecutive recurrent tumor samples of individual recurrent RM patients, whereas the width of the bars indicates the length within a chromosome/chromosome arm of the altered regions Different chromosomes (Chr) are depicted in different colors. P3: patient 3 (**a**); P4: patient 4 (**b**); P5: patient 5 (**c**); P6: patient 6 (**d**); P7: patient 7 (**e**); p and q indicate the long and short arm of each involved chromosome; del: deletion; del/gain: combined loss and gain of regions from the same chromosome; gain: short genetic segments gained; loss: one entire arm or complete chromosome lost; trisomy: one entire arm or complete chromosome gain.

**Figure 4 biology-13-00350-f004:**
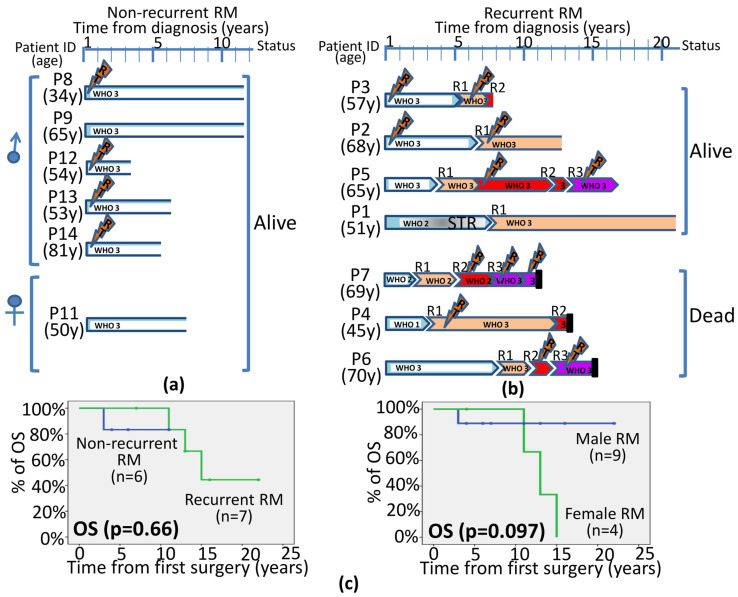
Detailed follow-up of RM patients (*n* = 13) with non-recurrent (**a**) and recurrent RM (**b**) for whom follow-up data was available and their overall survival (**c**) according to recurrence status time and gender. In the upper panels (**a**,**b**), primary specimens are labeled as blue (P), R1 specimens in orange, R2 in red and R3 as purple bars, including the corresponding WHO grades and type of surgical resection performed (gross total resection is depicted in white while the grey color indicates subtotal resection). Orange sparks indicate the moment when a patient received adjuvant radiotherapy. Open-ended bars are used for patients that remain alive while black-ended boxes indicate those that have died.

**Table 1 biology-13-00350-t001:** Demographics, clinical, histopathological and immunohistochemical characteristics of non-recurrent versus recurrent RM patients.

Disease Features	Diagnosis		*p*-Value	Recurrences of RM			*p*-Value
	Non-Recurrent (*n* = 8)	Recurrent (*n* = 7) ^Ө^		R1 (*n* = 7)	R2 ^#^ (*n* = 4)	R3 (*n* = 3)	
Age * (range)	59 (34–81)	61 (45–69)	0.9	69 (48–75)	72 (58–74)	78 (73–79)	0.06
Sex							
Man (*n* = 10)	75%	57%		57%	40%	33%	
Woman (*n* = 5)	25%	43%	0.4	43%	60%	67%	0.7
% of rhabdoid cells							
<50% (*n* = 7; 4 m & 3 w)	25%	71%		43%	75%	33%	
>50% (*n* = 8; 6 m & 2 w)	75%	29%	0.07	57%	25%	67%	0.5
WHO grade							
1	0	14%		0%	0%	0%	
2	0	72%	0.003	14%	25%	0%	0.6
3	100%	14%		86%	75%	100%	
*BAP1* expression (*n* = 26) ^&^	100%	33%	0.02	71%	100%	100%	0.3

R1: first recurrence; R2 second recurrence; R3: third recurrence; Ө: one primary specimen belonging to a patient who suffered 3 relapses was discarded from the *BAP1* and genetic analyses due to poor sample quality after decalcification; #: clinical data of one additional second recurrence was available, but not histopathological data, because the patient was treated with bevacizumab in the absence of surgery or radiotherapy; *: age expressed as median and range in years; &: total of tumors analyzed included 14 primary RM samples (7 non-recurrent and 6 recurrent primary tumors) and another 14 samples corresponded to first (*n* = 7), second (*n* = 4) or third (*n* = 3) recurrent tumor samples.

**Table 2 biology-13-00350-t002:** Treatment administered and outcome of non-recurrent versus recurrent RM patients.

Follow-Up Features		Diagnosis		*p*-Value	RM Recurrences			*p*-Value
		Non-Recurrent RM (*n* = 8)	Recurrent RM (*n* = 7)		R1 (*n* = 7)	R2 (*n* = 4) ^$^	R3 (*n* = 3)	
Treatment								
Surgery:	% GTR	88%	86%	0.9	100%	50%	0%	0.03
	% STR	12%	14%		0%	50%	100%	
% Radiotherapy		50%	29%	0.4	43%	50%	67%	0.2
Follow up (years) *		6 (3–11)	13 (7–22)	0.008	5 (2–9)	3 (1–9)	3 (<1–5)	0.01
Time to recurrence #		-	-	-	68 (20–114)	33 (21–118)	53 (9–71)	0.2
% of deaths		14% ^ɛ^	43%	0.5	0%	20%	67%	0.05

$: at the second relapse one patient (P3) did not undergo surgery or radiotherapy (*n* = 4) but it was included in survival data analyses (*n* = 5); * results expressed as median (range) in years or #: months; ɛ: one patient died around the time of surgery in the non-recurrent tumor subset.

**Table 3 biology-13-00350-t003:** Copy number alterations (losses and gains) detected in primary tumor samples from non-recurrent vs. recurrent RM.

CNA		Chr Location [N. of Genes]	RM			
Size	Type		Non-Recurrent (*n* = 8)		Recurrent (*n* = 6) *	
			Frequency	q-Value	Frequency	q-Value
Broad	Loss	1p [2121]	2/8 (25%)	-	4/6 (67%)	0.004
		14q [1341]	1/8 (12%)	-	4/6 (67%)	0.004
		18p [143]	1/8 (12%)	-	4/6 (67%)	0.004
		18q [446]	1/8 (12%)	-	4/6 (67%)	0.004
		19p [995]	5/8 (62%)	<0.0001	1/6 (17%)	-
		19q [1709]	4/8 (50%)	0.003	0/6 (0%)	-
		21p [13]	0/8 (0%)	-	3/6 (50%)	0.03
		22q [921]	8/8 (100%)	<0.0001	6/6 (100%)	<0.0001
	Gain	20p [355]	3/8 (38%)	0.04	1/6 (17%)	-
		21p [13]	3/8 (38%)	0.0006	1/6 (17%)	-
		21q [509]	3/8 (38%)	0.03	1/6 (17%)	-
Focal	Gain	17q22 [1]	0/8 (0%)	-	4/6 (67%)	0.03

CNA: copy number alteration; RM: rhabdoid meningioma; Chr: chromosome; *: one sample discarded because of poor DNA quality.

**Table 4 biology-13-00350-t004:** Comparison between the specific chromosomal changes found in paired diagnostic and first recurrent RM tumors including those genetic changes acquired.

CNA		Chr Location [N. of Genes]	Diagnosis		First Relapse	
Size	Type		Frequency	q-Value	Frequency	q-Value
Broad	Loss	1p [2121]	4/6 (67%)	0.004	4/6 (67%)	<0.001
		6q [839]	3/6 (50%)	0.07	3/6 (50%)	0.051
		14q [1341]	4/6 (67%)	0.004	4/6 (67%)	0.001
		16q [702]	2/6 (33%)	-	3/6 (50%)	0.025
		18p [143]	4/6 (67%)	0.004	4/6 (67%)	0.007
		18q [446]	4/6 (67%)	0.004	4/6 (67%)	0.006
		19p [995]	1/6 (17%)	-	3/6 (50%)	0.047
		21p [13]	3/6 (50%)	0.027	3/6 (50%)	0.045
		22q [921]	6/6 (100%)	<0.001	6/6 (100%)	<0.001
	Gain	1q [1955]	3/6 (50%)	-	4/6 (67%)	0.002
		17q [1592]	3/6 (50%)	-	4/6 (67%)	0.003
Focal	Gain	17q22 [1] (55,403,923–57,333,858)	4/6 (67%)	0.031	0/6 (0%)	0/6 (0%)

Chr: chromosome; N: number of genes in brackets; p: the short chromosome arm; q: the long chromosome arm.

## Data Availability

The data will be publicly archived and available upon request.

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
