# Peer review of "Paired Primary and Recurrent Rhabdoid Meningiomas: Cytogenetic Alterations, *BAP1* Gene Expression Profile and Patient Outcome"

_biology, 2024, doi:10.3390/biology13050350_

Round 1

Reviewer 1 Report

Comments and Suggestions for Authors

Comments on the Quality of English Language

Author Response

REVIEWER #1:

Comment 1. “Clarify patient and tumor selection criteria.”

Answer to comment 1.- Patient and tumor selection criteria are now clearly specified (as per the recommendation of the reviewer) in the material and methods section of the revised version of the manuscript.

Comment 2. “Expand discussion on non-significant statistical findings and their applications.”

Answer to comment 2.- Discussion has been expanded in the revised version of the manuscript on non-significant statistical findings and its applications.

Comment 3. “Discuss potential biases and limitations more thoroughly.”

Answer to comment 3.- The potential biases and limitations of the study are now discussed more thoroughly in a new sentence added in the text of the discussion section of the revised manuscript, in line with the recommendation of the reviewer.

Comment 4. Consider simplifying or further explaining dense figures and technical language.

Answer to comment 4.- Figure 3 and its legend, as well as the legends of all other figures have been edited for easier reading and understanding and technical language is now avoided.

Final comment. “The paper is fundamentally sound and contributes valuable......... for researchers and clinicians in the field.”

Answer to final comment.- We thank the reviewer for the overall positive final comment about the paper contents and we have nothing to add in this regard.

Reviewer 2 Report

Comments and Suggestions for Authors

This manuscript (biology-2952870) retrospectively the different types of tumors and revealed the discrepancy of chromosome losses in different groups. However, limited by small sample sizes, some conclusions show no statistically significant difference. So, I recommend this manuscript be rejected by biology.

Here are some additional questions and suggestions:

1.       All histological and immunohistochemical studies should be added scale bars.

2.       The Mann-Whitney u-test is used when the data does not have a normal distribution. So, you should confirm whether the sample conforms to a normal distribution or not before doing a u-test.

Author Response

REVIEWER #2:

Comment 1.All histological and immunohistochemical studies should be added scale bars.

Answer to comment 1.- Scale bars have been added in Figure 1, and its legend, in line with the recommendation of the reviewer.

Comment 2. “The Mann-Whitney U test is used when the data does not have a normal distribution. So, you should confirm whether the sample conforms to a normal distribution or not before doing a u-test.”

Answer to comment 2.- The distribution of the data was checked before the statistical tests for group comparisons were applied for significance analyses. This is now specified in the text of the methods section on statistical methods

Reviewer 3 Report

Comments and Suggestions for Authors

The authors Ruiz et al., depicted a novel biomarker involvement in the Rhabdoid meningioma. The study was well-designed and the results were clear. The paper is of atmost interest to the scientific community studying meningiomas.

There are a couple of minor corrections needed:

1. Figure 1: It would be clear to point out the arrows at the BAP1 expression in the images or for example: describe that BAP1 is depicted in blue and we see decrease in the expression across the samples. 

2. Figure 3 is a little blur and it would be good to explain the Sankey flow diagrams in detail.

Other than that, I think the paper can be accepted. 

Comments on the Quality of English Language

Minor grammar check required. No major corrections needed. 

Author Response

REVIEWER #3:

Comment 1.Figure 1: it would be clear to point out the arrows at the BAP1 expression in the images or for example: describe that BAP1 is depicted in blue and we see decrease in the expression across the samples.

Answer to comment 1.- We appreciate the reviewer's comments about Figure 1 that has been edited (scale bars added in it), in line with the recommendation of the reviewer.

Comment 2. “Figure 3 is a little blur and it would be good to explain the Sankey flow diagrams in detail.”

Answer to comment 2.- Figure 3 has been edited and the test of its legend expanded to explain in more detail the observations contained in it, in line with the recommendation of the reviewer.

Round 2

Reviewer 2 Report

Comments and Suggestions for Authors

The manuscript (biology-2952870) has been improved. As a result, I give the suggestion of accepting this manuscript.